# Stem Cell-Based Regeneration and Restoration for Retinal Ganglion Cell: Recent Advancements and Current Challenges

**DOI:** 10.3390/biom11070987

**Published:** 2021-07-05

**Authors:** Jingxue Zhang, Shen Wu, Zi-Bing Jin, Ningli Wang

**Affiliations:** 1Beijing Institute of Ophthalmology, Beijing Tongren Eye Center, Beijing Tongren Hospital, Capital Medical University, Beijing Ophthalmology & Visual Sciences Key Laboratory, Beijing 100730, China; jingxuezh@ccmu.edu.cn (J.Z.); kiddshenx@126.com (S.W.); 2Beijing Institute of Brain Disorders, Collaborative Innovation Center for Brain Disorders, Capital Medical University, Beijing 100069, China

**Keywords:** stem cells, glaucoma, retinal ganglion cell, regeneration, cell replacement

## Abstract

Glaucoma is a group of irreversible blinding eye diseases characterized by the progressive loss of retinal ganglion cells (RGCs) and their axons. Currently, there is no effective method to fundamentally resolve the issue of RGC degeneration. Recent advances have revealed that visual function recovery could be achieved with stem cell-based therapy by replacing damaged RGCs with cell transplantation, providing nutritional factors for damaged RGCs, and supplying healthy mitochondria and other cellular components to exert neuroprotective effects and mediate transdifferentiation of autologous retinal stem cells to accomplish endogenous regeneration of RGC. This article reviews the recent research progress in the above-mentioned fields, including the breakthroughs in the fields of in vivo transdifferentiation of retinal endogenous stem cells and reversal of the RGC aging phenotype, and discusses the obstacles in the clinical translation of the stem cell therapy.

## 1. Introduction

Glaucoma is a heterogeneous group of chronic neurodegenerative disorders characterized by relatively selective, progressive damage to the retinal ganglion cells (RGCs) and their axons, which leads to axon loss and visual field alterations. The heterogeneity of the disease makes its treatment difficult [1,2]. It is the most common cause of irreversible blindness worldwide [3], currently affecting almost 80 million people. By 2010, 1 out of 15 blind people was blind due to glaucoma, and 1 of 45 visually impaired people was visually impaired due to glaucoma [4].

The pathologically elevated intraocular pressure (IOP) is the main but not the only risk factor for the progression of the glaucoma damage. However, treatments aimed at decreasing IOP are ineffective in many cases, which might be explained in a few ways. First, the values obtained through IOP measurement cannot reflect the real physiological state, which affects the setting of target IOP. It is important to keep in mind that IOP is not a static parameter but undergoes dynamic changes; thus, it is of great need to further develop measurement equipment that allows long-term, continuous monitoring of IOP [5]. Second, though the surgical intervention is considered the most effective procedure for lowering IOP in uncontrolled glaucoma with ocular hypertension, it is not available to the vast majority of patients to reach an acceptable IOP target, such as Refractory glaucoma [6]. In addition, even if the intraocular pressure is effectively controlled, 15–37% of glaucoma patients still experience persistent RGC loss and progressive optic nerve damage [7,8]. Therefore, an effective treatment that focuses on the RGCs and optic nerve regeneration and protection is urgently needed.

The unlimited proliferation ability and multi-directional differentiation potential of stem cells have made cell replacement therapy a promising therapeutic option. The use of stem cell replacement therapy for corneal blindness, macular degeneration, and retinitis pigmentosa has become the hot spot of the relevant research, and globally, there are a number of ongoing clinical trials on the replacement of corneal cells and retinal pigment epithelial cells [9]. At present, a breakthrough on the replacement of photoreceptor cells and RGCs is yet to be made. It is expected that the progress on the replacement of photoreceptor cells might come through sooner, as photoreceptors can directly sense external light signal input, and their target sites are located in inner retinal neurons, within a short distance. However, the regeneration of RGCs requires sufficient synaptic integration of stem cells in the inner layer of the host retina and the development of long-distance axons that project to the brain and accurately form effective synaptic connections with their appropriate targets, such as dorsal lateral geniculate nucleus, suprachiasmatic nucleus and superior colliculus, thereby completing the signal transmission. Given its challenging nature, this process has been considered as the “last mile” of the application of stem cells in visual neuro-regeneration. This review discusses the recent development of stem cell-based RGC regeneration and replacement, including potential cell sources for RGC replacement and the main strategies for stem cell-based RGC regeneration and protection, and the challenges that we need to overcome before the stem cell-based therapy can be translated to clinical settings.

## 2. Source Cells for RGC Replacement

As the RGCs of adult mammals and humans lack the ability to self-regenerate and their self-repairment ability is inhibited by certain signals, the source cells for RGC replacement mainly come from the induced differentiation of retinal progenitor cells or stem cells. Due to the limited source of retinal progenitor cells, many studies have focused on the exploration of technology for the induced differentiation of stem cells into RGCs, and a series of breakthroughs have been made in the past decade.

Recently, we described in detail the various differentiation induction systems currently used in RGC replacement research [10,11]. Generally, the research into RGC induction in vitro mainly uses human or mouse embryonic stem cells (ESCs) and induced pluripotent stem cells (iPSCs). The induction simulates the retinal development process by adding relevant compounds/factors or conducting gene interventions according to the time sequence so that various transcription factors that promote the formation of RGCs are regulated to better play their role. The target transcription factors include Pax6, Math5, and Notch, etc. The time length of induction and differentiation varies based on the types of source cells and induction methods, ranging from 10 to 60 days. There is also a great difference in the induction efficiency, ranging from 0.3% to 50%, which may correspond to the proportion of RGCs in different stages of normal retinal development. Teotia et al. [12] added relevant factors to initiate differentiation corresponding to the three stages (initiation, differentiation, maturation) of the RGC development and achieved a differentiation induction efficiency of 53.67% in 15 days, making their method a promising induction system.

In the past decade, the advent of the 3D retinal organoid has greatly boosted the differentiation of various retinal neurons, including RGCs [13,14,15,16,17,18,19,20]. As the three-dimensional induction model has a greater resemblance to the in vivo developmental environment and the spatial structure of retinal cells, the state of the obtained target cells is relatively mature; therefore, it has become the main system for obtaining RGCs through in vitro differentiation induction [21]. 

However, a series of technical difficulties are yet to be overcome in order to obtain sufficient functional RGCs for in vivo transplantation. Firstly, purification and enrichment are essential to ensuring the proper functioning of cells and in avoiding tumor formation after transplantation. Currently, the main RGC purification methods include immumopanning, fluorescence-activated cell sorting (FACS), and magnetically activated cell sorting (MACS). By exploring the RGC-specific marker THY1, RGCs can be better isolated and purified from other cells [22,23], and RGCs have been successfully isolated from 3D retinal organoid differentiated from hESC [24]. Nevertheless, the surface recognition antigen THY1 is not only expressed on the surface of RGCs, but also on other types of ganglion cells. Hence, if there are other types of ganglion cells during the differentiation of stem cells, the cell purity of RGCs cannot be guaranteed. For that reason, it is of great importance to explore highly specific surface markers of RGCs, such as the recently discovered surface antigens CD184 and CD171, both of which are considered to have the application potential for the purification of RGCs and RGC precursors [25]. 

In addition, it is necessary to determine which developmental stage of RGC is more suitable for in vivo transplantation. In the study of photoreceptor cell replacement [26], it was found that the transplantation of photoreceptor precursor cells postnatal for 4–8 days makes it easier for the transplanted cells to integrate with the host retina and significantly improves visual function. However, it cannot be simply assumed that an earlier developmental stage of donor cells always guarantees a better therapeutic effect. Some studies have suggested that retinal cells at the early developmental stage may lose the ability to differentiate into target cells after transplantation [27,28,29]. Therefore, at present, it can only be inferred that the developmental stage of transplanted cells has a great influence on the therapeutic effect. In addition, the issue of RGC heterogeneity should also be taken into consideration while investigating the most suitable RGC subtype for in vivo transplantation. RGC is divided into many subtypes, and more than 40 subtypes have been identified by recent studies. Different subtypes of RGC not only have different molecular characteristics and physiological functions but also respond differently to disease damage [30,31]. Therefore, transplantation of some glaucomatous damage-resistant RGC subtypes may improve or prolong the therapeutic effect.

Nonetheless, the broad-spectrum markers such as Brn3, Thy1, Pax6, Rbpms, and Isl1 that are widely used in current researches to identify RGCs derived from induced differentiation are not able to properly identify the development stage and subtypes of RGCs. The newly emerged single cell transcriptomics and single cell RNA-sequencing technology provided feasible solutions to the above two issues, as they can obtain the molecular characteristics of different RGC subgroups at different developmental stages from human tissues or 3D retinal organoids induced by hiPSC, providing relatively accurate identification markers for RGCs, then identifying the damage-resistant RGC subtypes [32,33,34].

## 3. Stem Cell Strategies in RGC Regeneration and Protection

### 3.1. Stem Cell Transplantation and Replacement

The stem cell-based therapy mainly replaces the lost neurons through cell transplantation and thus makes up for the disadvantage that human RGCs cannot repair themselves. However, after highly undifferentiated stem cells are implanted into the retina, they rarely successfully differentiate into targeted retinal neurons. For example, after iPSC is directly transplanted into the vitreous cavity or subretinal space of the glaucoma animal model, these cells mainly differentiate into oligodendrocytes spontaneously [35]. Thus, currently, in most research into cell replacement, the transplanted cells are initially induced into a specific cell lineage in vitro and are then transplanted for RGC replacement. The main types of transplanted cell include primary RGC [22,23], ESC-derived neural progenitor cells [36], the iPSC-derived RGC [37], RGC precursors derived from Müller cells [38], and RGC derived from spermatogonial stem cells [39]. 

Generally, the main methods of transplantation are subretinal space transplantation and vitreous cavity transplantation. The transplanted cells can survive for several days to several months and integrate into the host retina, with the ganglion cell layer (GCL) as the primary location of integration (Table 1). However, in these studies, the evaluation of visual function is rarely carried out. Divya et al. [36] transplanted ESC-NPC into the vitreous cavity of an NMDA-induced retinal injury rat model. The behavioral test with the Light Avoidance Experiment revealed a certain therapeutic effect on the improvement of visual function. Singhalet et al. [38] and Eastlake et al. [40] transplanted RGC precursors derived from Müller cells/human iPSC-Müller cells into rat models and found that the response of the electroretinography (ERG) increases. Most of the above studies did not describe the survival rate of transplanted cells in the host retina, the distribution of transplanted cells and the evaluable transplantation success rate. Recently, Oswald et al. [37] used a survival rate of >5% after donor cell transplantation as the indicator for a successful transplant and found that the miPSC/mESC-derived donor RGCs can survive for up to 12 months after transplantation with a success rate of over 65%, which is far higher than the 10% success rate of primary RGCs transplantation [22,23]. In addition, researchers have also carried out RGCs replacement research in some non-human primates. Chao et al. [41] transplanted hESC-retinal neurons into the subretinal space of squirrel monkeys and found that the axons of some transplanted cells extend along the host nerve fiber layer towards the optical nerve head and have the ability to integrate into the optic nerve bundle and project to the brain. These results indicate that RGC replacement therapy is feasible, and the transplanted cells have the ability to integrate with the host retina and the potential to form effective synaptic connections in the visual pathway.

### 3.2. Stem Cell-Mediated Neuroprotection

In addition to replacement, intraocular transplantation of certain stem cells also has neuroprotective effects on surviving RGCs, mainly mesenchymal stem cells (MSCs), and these cells also possess the merit of easy-to-obtain autologous cells for therapeutic applications (Table 2). In 2006, Yu S. et al. [48] transplanted bone marrow MSCs into the eyes of glaucoma animal models, and found that exogenous cells can be integrated into the GCL and IPL layers, and can effectively reduce the loss of RGC, suggesting that MSCs have application potential in the treatment of glaucoma. Afterward, researchers used MSCs from different sources (bone marrow, dental pulp) to study the treatment of glaucoma with animal models, and elucidated the protective role of MSCs in glaucomatous optic nerve damage [48,49,50,51,52,53]. However, in their research, they also found that although the transplanted MSCs can be integrated into the GCL layer, they hardly differentiate into RGC, which suggested that the therapeutic effect of MSCs after transplantation mainly derives from the neuroprotective effect of neurotrophic factors secreted by MSCs on neurons and their regulatory effect on the microenvironment.

MSCs can secrete a variety of cytokines, including brain-derived neurotrophic factor (BDNF), ciliary neurotrophic factor (CNTF), glial cell line-derived neurotrophic factor (GDNF), and basic fibroblast growth factor (BFGF), among which, BDNF has been proven to be an essential and effective protective factor for glaucoma optic nerve damage [54]. Researchers further constructed MSCs that stably overexpress BDNF and used these cells for intraocular transplantation experiments in glaucoma animal models. The results uncovered that the neuroprotective effect of MSCs that stably overexpress BNDF on damaged RGCs was significantly enhanced [55,56]. So far, a variety of strategies for the application of neurotrophic factors in the treatment of glaucoma and other neurodegenerative diseases have been proposed. Given the goals of this type of treatment, the advantage of stem cell transplantation lies in the long-term release of multiple therapeutic factors through a single injection, thereby reducing the risk of infection and bleeding caused by the intraocular injection, as well as the burden of patient compliance.

In recent years, exosomes secreted by MSCs have gradually gained attention in MSC-mediated neuroprotection. Exosomes are endocytic structures composed of proteins, lipids and mRNA, which can regulate the translation of proteins in target cells when delivered to target cells [57]. Intraocular transplantation of BMSC-derived exosomes extracted in vitro can effectively protect RGCs and enhance the visual function of glaucoma animal models [58,59,60]. In the further study of the effective components of exosomes, it was found that miRNAs in exosomes may be the key components for neuroprotection [58], which provides a new approach for the neuroprotective treatment of glaucoma.

### 3.3. Endogenous Transdifferentiation and Nerve Repair

Theoretically, activating the retinal endogenous regeneration and repair is the most direct treatment for RGC loss (Table 3). The retinal Müller cells in fish and amphibians have a high regeneration ability and can even regenerate the entire retinal tissue [64]. Unfortunately, this ability is greatly inhibited in mammals. Previous studies have confirmed that Müller cells can be induced to differentiate into RGCs through the inhibition of the Notch signaling pathway in vitro [42,65]. However, most of the new cells obtained by direct transdifferentiation in vivo are amacrine rather than RGCs [21]. Nonetheless, recently, an exciting breakthrough in the transdifferentiation of Müller cells into RGCs in vivo has been made. Yang’s group [66] developed RNA targeting CRISPR system CasRx to specifically down-regulate the expression of polypyrimidine tract binding protein 1 (Ptbp1) in Müller cells, realizing the transdifferentiation of Müller cells into RGCs in vivo, and the transdifferentiated RGCs produced a corresponding electrical signal response to light stimulation and establishes central projections to dorsal lateral geniculate nucleus and superior colliculus, which effectively restored the light sensitivity in the mouse model. However, in the clinical translation of this technique, it is crucial to take notice of the fact that the CRISPR-based methods referred to in this study used specific transgenic mice to limit CRISPR targeting exclusively to the Muller glia. This is predictably a great impediment in the therapy of (arguably non-transgenic) humans, where the system will promiscuously transdifferentiate every glial element to a variety of neural elements in the retina. Recently, Xiang’s group [67] reported that overexpression of Math5 and Brn3 are sufficient to reprogram mature mouse Müller cells into RGCs with exceedingly high efficiency. Moreover, the axons of the newborn RGCs transdifferentiated from Müller cells can pass through the optic chiasm and project to the corresponding brain areas, including image-forming visual pathways and non-image-forming visual pathways, which indicated that RGCs derived from differentiation already have a variety of subtype characteristics.

Except for the transdifferentiation of other types of cells in the retina, recent research work on reversing the senescence phenotype of RGC is also quite enlightening [68]. The researchers first clarified the related changes in retinal epigenetics caused by aging, and inspired by the construction of iPS cells, they proposed that the compatibility of transcription factors may change the epigenetics of cellular aging into a healthy state. Considering the severe tumorigenicity of Myc, the study finally used transcription factors Oct4, Sox2 and klf4 to intervene RGCs and found that it can significantly promote the rejuvenation of the aging RGC phenotype and the regeneration of axons. The team further discovered through the regulation of transcription factors in glaucoma animal models that the glaucomatous optic nerve damage can be delayed, which not only provides an effective neuro-repair strategy for RGC but also, and more importantly, confirms that RGC damage caused by glaucoma can be reversed and treated. However, as the transcription factors used in the study are still closely related to tumorigenesis, the safety issue should be prioritized in the clinical translation.

### 3.4. Cellular Materialtransfer

In 2016, Pearson et al. [69] discovered that there is a cellular component exchange between transplanted donor photoreceptors and host retinal cells, including fluorescein protein. However, whether such a mechanism exists in the transplantation of RGCs remains to be confirmed. Moreover, whether such a cellular component exchange is completed through vesicle transport, exosomes or direct cell–cell contact has not yet been determined. Recent research by our team [70] found, through living cell observation, that hiPS-derived MSC can “donate mitochondria and lysosomes” to a variety of ocular tissue cells (including corneal endothelium, retinal pigmented epithelium, and photoreceptors), and the transferred mitochondria can help three types of damaged ocular cells repair their mitochondrial function and effectively reduce apoptosis. The supply of healthy mitochondria by such hiPS-MSCs to the photoreceptor mainly relies on F-actin-based tunneling nanotubes (TNT) formed between donor cells and host cells. Recent studies have shown that the majority of predicted integrated photoreceptors are the result of the cellular exchange, and that such an exchange is likely to be a more efficient approach to rescuing degenerative cells [71,72]. Zou et al. [73] found that transplanted C-Kit+ cells from hESC-derived retinal organoids have the potential to integrate into the ONL, though with a small proportion migrating into the inner retina, becoming inner retinal cells, or exchanging materials with inner retinal cells, including retinal ganglion cells. The above research indicates the diversity and complexity of communication between donor cells and host cells after transplantation.

## 4. Challenges Ahead and Perspective

There are still a few issues we need to take into consideration in the clinical translation of stem cell therapy for glaucoma.

First of all, the most common concern in stem cell replacement therapy has always been potential ethical issues, as well as the immune response and tumorigenicity brought about by allogeneic transplantation. Though hESC has been studied at the early stage of the RGC replacement therapy, its clinical translation has been hindered by the above-mentioned issues. Since hiPSC can be derived autogenously, the ethical issues and immunological rejection are much easier to solve. However, if it is to be used in clinical practice, the cost of constructing autogenous iPSCs for each patient would be relatively high, and thus, it would not be easy to promote such a strategy clinically. In recent years, researchers have been focusing on building an iPSC cell bank for HLA matching, which can be used for transplantation directly after matching, so as to promote the industrialization of cell transplantation. Recently, Kyoto University used gene-editing technology to produce HLA-I-deficient iPS cells to further produce HLA-deficient platelets. In addition, they confirmed with cell experiments and animal model experiments that iPSC-derived platelets depleted of HLA would not be attacked by anti-HLA antibodies and NK cells. This study proves that iPSC-derived platelets depleted of HLA can be used as a “universal” broad-spectrum preparation for blood transfusion, further reducing the cost of scale preparation of multiple iPSC-derived platelets depleted of HLA subtypes [74], and also provides a new perspective for the construction of such “universal” RGCs from HLA-I-deficient hiPSC sources. 

Secondly, genetic factors also play a role in the development of glaucoma; thus, the constructed autologous iPS cells still carry the pathogenic genetic information. If RGC is the target of the pathogenic gene, RGC induced by iPS will still carry damaging factors, and thus, it may be necessary to consider in vitro gene repair prior to in vivo transplantation. Previously, our team constructed the iPSC of retinitis pigmentosa patients, and the iPSC was differentiated into 3D retinal organoids carrying the pathogenic gene information. Afterwards, we used CRISPR/Cas9 gene-editing technology to repair the mutation and found that the occurrence of retinal degeneration was successfully prevented in terms of the gene expression profile, protein level, and electrophysiological characteristics of the cell, validating the proof-of-concept of curing the cells before curing the disease [75]. However, it must be borne in mind that the morbid state of the patients did not change, and the newly transplanted or endogenously regenerated RGCs are still vulnerable to the pathophysiological state. Therefore, even after the completion of stem cell therapy, glaucoma patients may still need to receive life-long intraocular pressure, lowering therapy and neuroprotection therapy.

In addition, the integration of the transplanted cells and the host retina needs to be further improved. After intravitreal or subretinal transplantation, the migration of the transplanted cells to the ganglion cell layer and its integration with the host retina are seriously affected by the physical barrier of the internal limiting membrane and pathological reactive gliosis. Researchers have discovered that aminoadipic acid (AAA), a drug that selectively inhibits glial cell activation, can significantly improve the migration and integration of cells after transplantation. In the study, the cells were injected into the vitreous cavity of normal mice and mice with glial response inhibition (AAA treatment), and it was found that more transplanted cells were integrated into the host retina of AAA-treated mice than in the control group [76]. The application of tissue engineering has also provided a new solution. Tissue engineering allows targeted transplantation in the form of cell sheet structure, which is easier to locate and integrate. Furthermore, the material constitutes a better carrier of supplementary substances for the synergistic effect. Li et al. [77] generated RGCs from hiPSCs-3D neural retina and then seeded these RGCs on a biodegradable poly (lactic-co-glycolic acid) (PLGA) scaffold to create an engineered RGC-scaffold biomaterial. Moreover, they transplanted into the intraocular environment of rabbits and rhesus monkeys, providing a technique to produce the engineered human RGC-scaffold biomaterial. In the clinical translation of glaucoma cell replacement therapy, it is necessary to explore more effective targets and methods to deal with the barrier of the integration of transplanted cells and host retina.

Another challenge lies in the regeneration and effective projection of RGC axons. Strictly speaking, the functional replacement of RGC requires the completion of the synapse integration in the retina, and also, the RGC axons need to extend along the optic nerve to the target brain areas to form effective signal projection. Although axon regeneration in the central nervous system is hindered by many factors, a series of studies have confirmed that through the regulation of certain transcription factors and signaling pathways, RGC axon regeneration and projection to brain areas can be induced [78,79,80,81]. For example, knocking out the transcription factor Klf4 in RGC can significantly promote the regeneration of axons after an optic nerve crush [80]. There are also researchers combining a variety of interventions to promote the axon regeneration effect [81]: (1) intraocular injection of zymosan to trigger the inflammatory response; (2) intraocular injection of cAMP analogs to enhance RGC axon regeneration signals; (3) knockout of the transcription factor PTEN. This combination treatment achieved full-length regeneration of RGC axons in mice with optic nerve damage. The regenerated RGC axons projected to the corresponding brain areas including the lateral geniculate and superior colliculus, and part of the visual function was restored. Another group found that combining neural activity with activation of mTOR can enhance axon regeneration and re-establish the correct patterns of connectivity [82]. In addition, myelination plays an important role in the rapid signal transmission and the proper functioning of axons. However, the regenerated RGC axons obtained by the current methods of promoting optic nerve regeneration mostly remain unmyelinated and ineffective at supporting visual functions. Wang et al. [83] found that optic nerve injury induces GPR17 expression in oligodendrocyte precursor cells (OPC) and chronically activates microglia to suppress the differentiation of OPC into mature myelination-competent oligodendrocytes. Co-manipulation of GPR17 and microglia promotes the extensive myelination of regenerated axons. These results suggest that we need to consider adding additional interventions to promote axon regeneration during the RGC replacement therapy.

Despite the above-mentioned challenges, there is no doubt that stem cell therapy has become the most promising therapeutic approach for the regeneration and protection of RGCs in advanced glaucoma. It could be expected that with the breakthrough in, and combined application of, biotechnologies such as single-cell omics, gene editing, tissue engineering and nanotechnology, the use of stem cells to protect, regenerate and replace damaged RGCs will play an essential role in helping glaucoma patients to regain their sight.

## Figures and Tables

**Table 1 biomolecules-11-00987-t001:** Cell transplantation for RGC replacement.

Donor Cells	Source of Donor Cells	Animal Model	Position of Transplantation	Does (Cells/Eye)	Layers of Integration	Cell Survival Time (Observation Period)	Survival Numbers/Rates of Transplanted Cell	Immunosuppressant	Functioal Evaluation	Ref.
**Müller-RSC (Aton7 overexpressed)**	rat	Laser induced hypertension rat	vitreous	50,000	INL-GCL	14 days	N/A	N/A	N/A	[42]
**Müller-RSC-RGC**	rat	Laser induced hypertension rat	Subretinal	50,000	GCL	14 days	N/A	N/A	N/A	[43]
**Müller-RGC precursor**	Human	NMDA induced retina injury Rat	Intravitreal	40,000	GCL	4 weeks	N/A	cyclosporine A/prednisolone/azathioprine	ERG preserved	[38]
**NPCs-RGC**	rat	Optic Nerve Axotomy rat	Intravitreal	50,000	GCL	1 week	1600 (24 h)/600 (1 week)	N/A	N/A	[22]
**iPSC/ESC -RGC**	mouse	WT/Microbeads induced hypertension/ NMDA induced retina injury mouse	Intravitreal	20,000	GCL	48 weeks/2 weeks	1%	N/A	N/A	[37]
**ESC-RPC**	mouse	NMDA induced retina injury mouse	Intravitreal	Unknown	GCL	8 weeks	N/A	N/A	N/A	[44]
**SSC-ESC-RGC**	mouse	NMDA induced retina injury mouse	Intravitreal	10,000	GCL	10 days	N/A	N/A	N/A	[39]
**ESC-NPC**	mouse	NMDA induced retina injury mouse/DBA2J	Intravitreal	1,000,000	GCL	8 weeks	N/A	N/A	Visual acuity increased; Improve Visual function (Light Avoidance Experiment) /both Negative in DBA2J	[36]
**iPSC-Müller**	human	NMDA induced retina injury rat	Intravitreal	100,000	GCL	4 weeks	N/A	Azathioprine/prednisolone/cyclosporin A	ERG preserved	[40]
**ESC-RPC**	human	WT mouse	Subretinal and epiretinal	50,000	GCL/INL, ONL	12 weeks	N/A	N/A	N/A	[45]
**ESC-RGC**	human	WT rat	Intravitreal	50,000	GCL	7 days	19–25/mm^2^ retina wholemount	N/A	N/A	[46]
**ESC-Retinal Neurons**	human	Squirrel monkey	submacular space	1,000,000	GCL, INL	12 weeks	N/A	N/A	N/A	[41]
**ESC-NP**	human	WT sabra rat	Intravitreal/ Subretinal	60,000–100,000	IPL	4 weeks/ 8–16 weeks	55,611/197,481	cyclosporine A	N/A	[47]

SC: Stem cell; ES: Embryonic stem cell; iPS: Induced pluripotent stem cell; RSC: Retinal stem cell; RPC: Retinal progenitor cell; NPC: Neural progenitor cells; NP: neural precursors; SSC: spermatogonial stem cells; GCL: Ganglion cell layer; INL: Inner nuclear layer; IPL: Inner plexiform layer; ONL: Outer nuclear layer; LAE: Light Avoidance Experiment.

**Table 2 biomolecules-11-00987-t002:** Cell transplantation for RGC neuroprotection.

Type of SC	Sources	Animal Model	Position of Transplantation	Does Cells/Eye	Integration	Observation Period (Survival Rates/Numbers of Transplanted Cell)	Functional Factors	Immunosuppressant	Positive Results of Functional Evaluation	Ref
**Lineage negative cells from bone marrow**	mouse	NMDA induced retina injury mouse	Intravitreal	100,000	GCL, INL	21 days	CNTF, BDNF, GDNF	N/A	N/A	[61]
**IGF1-NPC**	human	Microbeads induced hypertension mouse	Intravitreal	200,000	N/A	30 days	IGF1	N/A	Enhanced RGC survival Enhanced Axon survival	[62]
**BDNF-MSCs**	rat	Laser induced hypertension rat	Intravitreal	200,000	GCL	42 days (17.62% survival rate)	BDNF	N/A	ERG preserved Enhanced RGC survival	[55]
**BDNF-NPC**	N/A	optic nerve crush rat	Intravitreal	160,000	N/A	N/A	BDNF	N/A	Enhanced RGC survival	[63]
**NTF-MSCs**	human	optic nerve crush rat	Intravitreal	400,000	N/A	24 days (418 cells/10 slides)	BDNF, GDNF	Cyclosporine A	Enhanced RGC survival	[56]
**MSCs**	mouse	Normal saline anterior chamber perfusion mouse	Intravitreal	50,000	N/A	N/A	miR-21 / PDCD4	N/A	Enhanced RGC survival Inflammatory factors decreased Microglia inhibition	[49]
**BMSC**	rat	Episcleral vein cautery induced hypertension rat	Intravitreal	200,000	GCL	8 weeks (few transplanted cells)	CNTF, bFGF	N/A	Enhanced RGC survival	[48]
**MSCs**	Human	Intracameral injection of TGFβ1 induced hypertension rat	Intravitreal	150,000	N/A	N/A	N/A	N/A	RNFL preserved Enhanced RGC survival pSTR preserved	[50]
**BMSC**	unknown	Laser induced hypertension rat	Intravitreal	30,000/100,000	N/A	N/A	N/A	N/A	Enhanced RGC survival Improve Visual function (Water maze)	[51]
**BMSCs**	human	Human Retinal Explant	directly drop	5000	N/A	7 days	PDGF	N/A	Enhanced RGC survival Enhanced NeuN+ cells survival	[52]
**DPSC**	rat	optic nerve crush rat	Intravitreal	150,000	N/A	21 days	NGF, BDNF, NT-3	N/A	Enhanced RGC survival RNFL preserved Axon regeneration	[53]
**Exosomes derived from BMSC**	human	optic nerve crush rat	Intravitreal	3 × 10^9^	N/A	N/A	miRNA in exosomes	N/A	RNFL preserved pSTR preserved Axon regeneration	[58]
**Exosomes derived from MSCs**	human	Ischemia-Reperfusion rat	Intravitreal	4 × 10^6^	N/A	N/A	N/A	N/A	pSTR preserved Apoptosis decreased Inflammatory factors decreased	[59]
**Exosomes derived from BMSC**	human	DBA/2J mouse	Intravitreal	1 × 10^9^	N/A	N/A	N/A	N/A	Enhanced RGC survival pSTR preserved	[60]

BMSC: Bone Marrow-derived Stem/stromal Cells; DPSC: Dental pulp stem cells; MSC: Mesenchymal stem cells; NTF: Neurotrophic factor; NPC: Neural progenitor cell; SC: Stem cell; GCL: Ganglion cell layer; INL: Inner nuclear layer; pSTR: Positive scotopic threshold response; RNFL: Retinal nerve fiber layer.

**Table 3 biomolecules-11-00987-t003:** Endogenous transdifferentiation and rescue of RGC.

Factors	Method of Transfection	Animal Model	Efficiency	Mechanism	Function	Ref.
**math5, Brn3b/a/c overexpression**	AAV9	optic nerve crush mouse	92.8% Rbpms+ RGCs of infected Müller 67.9% Brn3a+ RGCs of infected Müller	reprogramming Müller to RGC	Connect to appropriate central targets, Exhibited typical neuronal electrophysiological properties Increase of RGCs Stronger VEP response	[67]
**OCT4, SOX2, KLF4 overexpression**	AAV2	optic nerve crush mouse Microbeads induced hypertension mouse	40% RGC infected	restores youthful DNA methylation patterns and transcriptomes of aging and injured RGC,	Promote RGC survival promote axon regeneration Restore visual function (optomotor response & PERG)	[68]
**Ptbp1 knockdown**	AAV-CasRx-ptbp1	NMDA Induced Retinal Injury mouse	more than half infected Müller cells in GCL expressed Brn3a and Rbpms	reprogramming Müller to RGC	Connect to appropriate central targets, vision-dependent behavior restored (Light Avoidance Experiment;)	[66]

VEP: Visual Evoked Potential; PERG: pattern electroretinogram.

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
