# Peer review of "Stem Cell-Based Regeneration and Restoration for Retinal Ganglion Cell: Recent Advancements and Current Challenges"

_biomolecules, 2021, doi:10.3390/biom11070987_

Round 1

Reviewer 1 Report

The paper is very interesting and well written.  However, I have only some minor concerns related to the introduction and to my point of view it needs to be improved. Indeed, it does not provide a complete overview about the glaucoma disease to the readers that are not glaucoma specialists.

Below I will be very grateful if authors could address shortly my suggestions:

1) Please add In the introduction section that glaucoma is an heterogeneous group of chronic neurodegenerative disorders characterized by a relatively selective, progressive damage to the retinal ganglion cells (RGCs) and their axons, which leads to axon loss and visual field alterations. The heterogenity of the disease makes its treatment difficult.

Please use the following recent reference:

  • doi: 10.3390/ijms22094323.

2) Please add some epidemiological data regarding regarding the impact of glaucoma. It is the most common cause of irreversible blindness worldwide, currently affecting almost 80 million people, or more than 1% of the global population. By 2010, 1 out of 15 blind people was blind due to glaucoma, and 1 of 45 visually impaired people was visually impaired due to glaucoma, highlighting the increasing global burden of glaucoma. In this way, the reader will understand why the topic that authors are discussing is an “hot topic” and of interest for the pubblic health.

Please use the following reference:

  • doi: 10.1177/1120672120960339.

3) Please explain better that IOP is the main but not the only risk factor for the progression of the glaucoma damage. Indeed, treatments aimed at decreasing IOP are ineffective in many cases. This could do also to the evidence that IOP is not a static parameter but undergoes dynamic changes. Please clarify this aspect in the Introduction, using the following reference:

  • doi: 10.1155/2019/9890831

4) Surgical intervention, considered the most effective procedure for lowering IOP in uncontrolled glaucoma with ocular hypertension. However, it is not available to the vast majority of patients and often multiple interventations could be required to reach an acceptable IOP target. Indeed, to date, surgery is unavailable to more than three-quarters of patients suffering from vision impairment due to cataract or uncorrected refractive error, conditions that are fully treatable surgically. Please discuss these aspects in the introduction or discussion sections. It will help readers to better understand the backgroud of the study and the aims of the paper. Additionally, as stated by authors, these aspects justify the issue why an effective treatment that focuses on the RGCs and optic nerve regeneration and protection is urgently needed. Please discuss these aspects already reported by Fiedorowicz M et al (2020).

5) Posarelli C et al (2020) reported an interesting study and stated as Refractory glaucoma still represents a challenge for ophthalmologists to manage intraocular pressure. Indeed, even in case of well-controlled IOP, some visual field defects progress anyway to the blindness. Please state more incisively this aspect in the Introduction.

I hope the authors will consider my suggestions useful to improve their already interesting paper.

Looking to review the manuscript once again.

Best regards,

Author Response

WE SINCERELY THANK THE REVIEWER FOR THE CONSTRUCTIVE COMMENTS.

Reviewer #1: The paper is very interesting and well written.  However, I have only some minor concerns related to the introduction and to my point of view it needs to be improved. Indeed, it does not provide a complete overview about the glaucoma disease to the readers that are not glaucoma specialists.

Below I will be very grateful if authors could address shortly my suggestions:

1) Please add In the introduction section that glaucoma is an heterogeneous group of chronic neurodegenerative disorders characterized by a relatively selective, progressive damage to the retinal ganglion cells (RGCs) and their axons, which leads to axon loss and visual field alterations. The heterogenity of the disease makes its treatment difficult.

Please use the following recent reference:  doi: 10.3390/ijms22094323.

Thank you for your suggestion. We've added the suggested content and reference in the introduction section [Reference# 2].

2) Please add some epidemiological data regarding regarding the impact of glaucoma. It is the most common cause of irreversible blindness worldwide, currently affecting almost 80 million people, or more than 1% of the global population. By 2010, 1 out of 15 blind people was blind due to glaucoma, and 1 of 45 visually impaired people was visually impaired due to glaucoma, highlighting the increasing global burden of glaucoma. In this way, the reader will understand why the topic that authors are discussing is an “hot topic” and of interest for the pubblic health.

Please use the following reference: doi: 10.1177/1120672120960339.

Thank you for your suggestion. We've added the suggested content and reference in the introduction section [Reference# 4].

3) Please explain better that IOP is the main but not the only risk factor for the progression of the glaucoma damage. Indeed, treatments aimed at decreasing IOP are ineffective in many cases. This could do also to the evidence that IOP is not a static parameter but undergoes dynamic changes. Please clarify this aspect in the Introduction, using the following reference:

 doi: 10.1155/2019/9890831

Thank you for your suggestion. We've further clarified it in the introduction section and added the suggested reference [Reference# 5].

4) Surgical intervention, considered the most effective procedure for lowering IOP in uncontrolled glaucoma with ocular hypertension. However, it is not available to the vast majority of patients and often multiple interventations could be required to reach an acceptable IOP target. Indeed, to date, surgery is unavailable to more than three-quarters of patients suffering from vision impairment due to cataract or uncorrected refractive error, conditions that are fully treatable surgically. Please discuss these aspects in the introduction or discussion sections. It will help readers to better understand the backgroud of the study and the aims of the paper. Additionally, as stated by authors, these aspects justify the issue why an effective treatment that focuses on the RGCs and optic nerve regeneration and protection is urgently needed. Please discuss these aspects already reported by Fiedorowicz M et al (2020).

Thank you for your suggestion. We've added further discussion on these aspects in the introduction section.

5) Posarelli C et al (2020) reported an interesting study and stated as Refractory glaucoma still represents a challenge for ophthalmologists to manage intraocular pressure. Indeed, even in case of well-controlled IOP, some visual field defects progress anyway to the blindness. Please state more incisively this aspect in the Introduction.

Thank you for your suggestion. We've added more incisive discussions on this aspect in the Introduction section and the following reference: doi: 10.3390/jcm9072039 [Reference# 6].

Reviewer 2 Report

The declared area of interest of this manuscript is the recent advancements and current challenges in the aspects of stem cell-based regeneration and restoration of RGCs. The title reflects its specificity, the manuscript has been bilaterally presented from both points of view and the other perhaps opposing concepts were also mentioned and broadly evaluated. However in its present form, this manuscript has several weaknesses, which have to be corrected before it can be considered for publication:

  1. Based on the cited references, if the survival rates of transplanted cells are low and the survival time is usually between 1-12 weeks, what is the real impact and the feasible future of this RGC replacement therapy?
  2. In this context, it would be necessary to show not only the transplanted cell numbers but the survival rates of these cells and/or efficiency of the used donor cells or type of SC in Tables 1 and 2.
  3. lines 47-48

“development of long-distance axons that project to the brain and accurately form effective synaptic connections with the neurons of visual cortex”

No, retinal ganglion cells do not give synapses to the visual cortex. They synapse to the (i) dorsal lateral geniculate nucleus of the thalamus (the vision multisynaptic pathway); (ii) the pretectum in the midbrain (pupillary reflex); suprachiasmatic nucleus (light level controlled autonomic functions) and (iv) superior colliculus (coordination of head and eye movements).

Only the axons from the postsynaptic cells in the dorsal lateral geniculate nucleus reach and synapse in the cortex. It has to be corrected!

  1. line 70

“et al.” is never used for a list of things. In modern English it is always used for persons in a list of authors. 

  1. Legend for Table 2:

“BMSC: Bone mesenchymal stem cells;” No, BMSC means Bone Marrow-derived Stem/stromal Cells. It has to be corrected!

6. The cellular material transfer section (Section 3.4) should be a bit more detailed and more relevant to RGC regeneration and restoration with more publications mentioned from this particular field.

7. Section 3.3

Line numbering is missing in this section. It is a problem because there are some issues here.

Again, the axons of the RGCs do not reach the visual cortex, as clearly stated in the original article under reference 62: “converted RGCs established central projections to dorsal lateral geniculate nucleus (dLGN) and superior colliculus (SC)”

Although in the Section 4 the authors clarified the issue (lines 87-88), it is still confusing for an uninitiated reader. It has to be corrected!

8. Another weakness of the paper is present here as well, the lack of critical evaluation of the advantages, the drawbacks and the human therapeutic potentials/risks of the different approaches. For instance, the CRISPR based methods referred to in this review used specific transgenic mice to limit CRISPR targeting exclusively to the Muller glia. This is predictably a great impediment in the therapy of (arguably non-transgenic) humans, where the system will promiscuously transdifferentiate every glial element to a variety of neural elements in the retina.

9. Altogether, although in Section 4 some of the challenges are mentioned, this review is primarily descriptive and comes off very short in technical details. There are quite a few breakthrough methodologies mentioned (e.g. AAA treatment to block glial activity; the complex treatments for axon neogenesis guidance; the RNA targeting CRISPR-CAS13; etc.). It leaves to the readers, however, to do their own searches to get any ideas what they really entail.

Author Response

WE SINCERELY THANK THE REVIEWER FOR THE CONSTRUCTIVE COMMENTS.

Reviewer #2: The declared area of interest of this manuscript is the recent advancements and current challenges in the aspects of stem cell-based regeneration and restoration of RGCs. The title reflects its specificity, the manuscript has been bilaterally presented from both points of view and the other perhaps opposing concepts were also mentioned and broadly evaluated. However in its present form, this manuscript has several weaknesses, which have to be corrected before it can be considered for publication:

Based on the cited references, if the survival rates of transplanted cells are low and the survival time is usually between 1-12 weeks, what is the real impact and the feasible future of this RGC replacement therapy?

Thank you very much for this excellent question. In fact, most of the previous studies focused on proposing new concepts or verifying feasibility, thus the observation period was not very long. In addition, the survival rate of cells after transplantation is closely related to the state of donor cells, the developmental stage of donor cells, and the state of the retinal microenvironment of the host. Therefore, we believe that the long-term survival, migration and integration of transplanted cells, the treatment of the disease will be achieved with the advancement of cell engineering technology and microenvironment modification technology and proper selection of cell transplantation window. As mentioned in the manuscript, a study published this year has confirmed that the transplanted cells can survive up to 12 months by optimizing the window period and method of transplantation [Reference# 37].

In this context, it would be necessary to show not only the transplanted cell numbers but the survival rates of these cells and/or efficiency of the used donor cells or type of SC in Tables 1 and 2.

Thank you very much for your kind suggestion. We strongly agree that the survival rate or efficiency of transplanted cells should be shown. Unfortunately, many published articles did not report the detailed data. We’ve added the data that were clearly reported in Table 1 and Table 2.

lines 47-48

“development of long-distance axons that project to the brain and accurately form effective synaptic connections with the neurons of visual cortex”

No, retinal ganglion cells do not give synapses to the visual cortex. They synapse to the (i) dorsal lateral geniculate nucleus of the thalamus (the vision multisynaptic pathway); (ii) the pretectum in the midbrain (pupillary reflex); suprachiasmatic nucleus (light level controlled autonomic functions) and (iv) superior colliculus (coordination of head and eye movements).

Only the axons from the postsynaptic cells in the dorsal lateral geniculate nucleus reach and synapse in the cortex. It has to be corrected!

Thank you very much for pointing out this error. This sentence has been corrected in the revised version.

line 70

“et al.” is never used for a list of things. In modern English it is always used for persons in a list of authors. 

Thank you very much for pointing out this error. This sentence has been revised in the revised version.

Legend for Table 2:

“BMSC: Bone mesenchymal stem cells;” No, BMSC means Bone Marrow-derived Stem/stromal Cells. It has to be corrected!

Thank you very much for pointing out this error. This word has been corrected in the revised version.

  1. The cellular material transfer section (Section 3.4) should be a bit more detailed and more relevant to RGC regeneration and restoration with more publications mentioned from this particular field.

Thank you for your suggestion. We've added more details and more relevant content, citing more publications.

  1. Section 3.3

Line numbering is missing in this section. It is a problem because there are some issues here.

Thank you for pointing this out and we are sorry for causing confusion. The Line number has been added in the revised version.

Again, the axons of the RGCs do not reach the visual cortex, as clearly stated in the original article under reference 62: “converted RGCs established central projections to dorsal lateral geniculate nucleus (dLGN) and superior colliculus (SC)”

Although in the Section 4 the authors clarified the issue (lines 87-88), it is still confusing for an uninitiated reader. It has to be corrected!

Thank you very much for pointing out this error. This sentence has been corrected in the revised version. Thank you.

  1. Another weakness of the paper is present here as well, the lack of critical evaluation of the advantages, the drawbacks and the human therapeutic potentials/risks of the different approaches. For instance, the CRISPR based methods referred to in this review used specific transgenic mice to limit CRISPR targeting exclusively to the Muller glia. This is predictably a great impediment in the therapy of (arguably non-transgenic) humans, where the system will promiscuously transdifferentiate every glial element to a variety of neural elements in the retina.

Thank you for your great suggestions. We've added more discussion on the advantages, drawbacks, and the human therapeutic potentials/risks of the different approaches in Section3.3.

  1. Altogether, although in Section 4 some of the challenges are mentioned, this review is primarily descriptive and comes off very short in technical details. There are quite a few breakthrough methodologies mentioned (e.g. AAA treatment to block glial activity; the complex treatments for axon neogenesis guidance; the RNA targeting CRISPR-CAS13; etc.). It leaves to the readers, however, to do their own searches to get any ideas what they really entail.

Thank you for your great suggestions. In Section 4, we’ve added descriptions on technological breakthroughs in the field of optic nerve regeneration, such as the technology of promoting optic nerve regeneration and forming correct projection by regulating RGC neural activity through visual stimulation, and the technology of promoting axon myelination by regulating microglia in optic nerve damage microenvironment.